# Effect of Cement Content on the Deformation Properties of Cemented Sand and Gravel Material

**Jie Yang [1,2], Xin Cai [2,3,*], Xing-Wen Guo [3] and Jin-Lei Zhao [4]**

[1] School of Transportation and Civil Engineering, Nantong University, Nantong 226019, China; Turtlesky@126.com

[2] College of Water Conservancy and Hydropower Engineering, Hohai University, Nanjing 210098, China

[3] College of Mechanics and Materials, Hohai University, Nanjing 210098, China; xwguo@hhu.edu.cn

[4] Jiangsu Surveying and Design Institute of Water Resources Co., Ltd., Yangzhou 225127, China; hhuzhjl@163.com

[*] Correspondence: xcai@hhu.edu.cn; Tel.: +86-138-1386-9666

**Abstract:** Knowing the deformation properties of cemented sand and gravel (CSG) material can help construct reasonable constitutive models for the material, which can be used to simulate the structural performance of various practical projects including CSG dams. In this study, to investigate the effect of cement content on the deformation properties of CSG material, we employ triaxial compressive tests for cement contents of 20, 40, 60, 80, and 100 kg/m$^3$ with a confining pressure range of 0.3–1.2 MPa, and theoretically analyze the results by the regression analysis prediction method. Here, we show that both cement content and confining pressure influence the deformation properties of CSG material: for an increase in cement content, the failure strain decreases and brittleness of CSG material increases; the initial modulus of the CSG material increased exponentially with increasing cement content or confining pressure; the peak volumetric strain and its corresponding axial strain increase linearly with increasing confining pressures, which decrease with increasing cement content; the initial tangent volumetric ratio can also be determined by the peak volumetric strain and its corresponding axial strain.

**Keywords:** cemented sand and gravel material; triaxial test; cement content; deformation property

---

## 1. Introduction

The cemented sand and gravel (CSG) dam, which is a type of cemented material dam, combines the merits of the concrete face rock-fill dam and roller compacted concrete dam, and provides the advantages of safety, economy, and eco-friendliness. In recent years, the number of CSG dam projects has increased. CSG dams are typically used in temporary projects, but they are becoming more popular in permanent constructions as well [1]. The security requirements for dams are constantly increasing. Therefore, it is necessary to systematically and thoroughly study the mechanical properties of CSG material to improve the accuracy of strain–stress forecasting results for CSG dams.

The mechanical properties of CSG material are the main basis for the constitutive model. At present, studies on the mechanical properties of CSG materials considering factors such as cement content, sand content, water-cement ratio, aggregate gradation, and curing age are mainly carried out using unconfined compression tests, flexural strength tests, and measurements of elastic modulus and Poisson's ratio [2–4]. Compared with roller compacted concrete, CSG material is only mixed with a small amount of cementing agent, and some of its mechanical properties are close to those of loose particles such as rockfill material. Therefore, triaxial tests can also be used to study the mechanical properties of CSG material. Lohani et al. analyzed the influence of curing age, water consumption,

cement content, density, and other mix design indexes on shear strength by triaxial shear tests of CSG material [5,6]. Haeri et al. analyzed the stress–strain characteristics of CSG material with different cement contents, and the following conclusions were obtained: the axial strain corresponding to the failure strength decreased with increase in cement content; when the cement content was less than 1.5%, CSG material would mainly shrink, and when the cement content was more than 1.5%, the material displayed obvious dilatancy; the specimen showed obvious strain softening; cohesive force, strength, and stiffness increased with increase in cement content [7–9]. Sun et al. successively carried out triaxial shear tests for the analysis of stress–strain curve characteristics for different cement contents [10]. Cai et al. studied mechanical properties, such as the failure strength and initial elastic modulus, under the influence of confining pressure [11]. Wu et al. carried out a large-scale triaxial shear test on CSG material and analyzed the influence of curing age on peak strength and stress–strain curve characteristics [12]. Fu et al. conducted static and dynamic triaxial tests on CSG material for different cement contents and studied the influence of cement content and confining pressure on the static and dynamic characteristics of CSG material [13].

The abovementioned studies on the mechanical properties of CSG material focused on the change trends of peak strength, stress–strain curve, and volumetric-strain–axial-strain curve under different confining pressures, curing ages, cement contents, or aggregate gradations. The cement content is thus one of the most important factors influencing CSG properties; most qualitative analyses of CSG materials are carried out with cement contents of 60 and 80 kg/m$^3$, which are commonly used in CSG dam engineering [13]. Yang et al. only considered the influence of different cement contents on the strength characteristics of CSG material and the corresponding strength expression [14]. However, the expressions for the deformation characteristics of CSG material under different cement contents, which can also indicate the relationship of deformation characteristics among rockfill material, roller compacted concrete, and CSG material, were not given.

In addition, for other cemented granular materials, some scholars have also studied mechanical properties [15–22]: Younes et al. considered the effect of cement content by conducting a triaxial test, and the results showed that the failure strength and its corresponding strain value were less than the corresponding results of the undrained tests; Liu et al. performed triaxial tests to study the mechanical properties of polyurethane foam adhesive reinforced rockfill materials; Wu et al. conducted an experimental study on the effect of aggregate gradation on dilatancy behavior and acoustic characteristics of cemented rockfill; the effect of fines on the mechanical properties of composite soil-stabilizer-stabilized gravel soil was analyzed by Zhao et al.; Zheng et al. analyzed the effects of morphological parameters of natural sand on the mechanical properties of engineered cementitious composites; Wu et al. performed a series of test studies to evaluate the potential impact of basalt fibers on the mechanical performance (compressive strength, flexural strength), fracture energy, and anti-shrinkage properties of cement-stabilized macadam; Farhad et al. conducted a comparative analysis between the conventional models in terms of evaluating mechanical properties of self-compacting and conventional concrete and developed new modulus of elasticity models, tensile strength models, and compressive stress–strain models for those materials; in addition, Farhad et al. investigated the compressive and splitting tensile strengths, modulus of elasticity and rupture, compressive stress–strain curve, and energy dissipated under compression at different curing ages for a control self-compacting concrete (SCC) mixture and three fiber-reinforced SCC containing steel, polypropylene, and hybrid (steel + polypropylene) fibers, and established their corresponding prediction models considering the effect of curing age. In these studies, only Younes et al. and Liu et al. qualitatively analyzed the effects of cement content and confining pressure on the deformation properties of cemented poorly graded sand–gravel mixtures, and polyurethane foam adhesive reinforced rockfill materials; however, they did not establish the quantitative expressions for quantitative indexes of these deformation properties [15,16].

Therefore, in this study, triaxial shear tests on CSG material with cement contents of 20, 40, 60, 80, and 100 kg/m$^3$ are designed and conducted. According to the test results, some expressions are derived

for the deformation characteristics for different cement contents, which can reveal the variation of deformation characteristics such as initial modulus, failure strains, peak volumetric strains, and their corresponding axial strains, based on the amount of cement content. The experimental study provides a theoretical basis for improving the accuracy of the strain–stress forecasting results of CSG material for applications in the CSG dam and other projects.

## 2. Materials and Methods

The CSG material used in this study includes cement, crushed stone, sand, and water. It should be noted that the usual maximum size of crushed stones for CSG material in actual projects is approximately 150 mm; however, owing to laboratory conditions, a size limit of 40 mm was set; the physical properties and composition of the crushed stone and the medium-sized sand with a fineness modulus of 2.48 are shown in Table 1. The physical properties and chemical composition of ordinary Portland cement (OPC; grade P.C. 32.5) from the Anhui Digang Hailuo Cement Co., Ltd. (Anhui, China) are shown in Table 2. Tap water was used in the CSG material formulation [14]. To explore the effect of cement content on the deformation properties of CSG material, different mix proportions of CSG materials for cement contents of 20, 40, 60, 80, and 100 kg/m$^3$ were considered and are given in Table 3, and the water–cement ratio was 1.0.

**Table 1.** Physical properties and composition of crushed stones and sand.

| Aggregate Type | Specific Gravity | Bulk Density (kg/m$^3$) | Water Content | Clay Content |
|---|---|---|---|---|
| Crushed stone | 2.71 | 1650 | 0.01% | 0.01% |
| Sand | 2.62 | 1450 | 0.01% | 0.01% |

**Table 2.** Physical properties and chemical composition of the cement.

| The Fineness | The Content of SO$_3$ | The Content of MgO |
|---|---|---|
| 2.26% | 2.56% | 1.78% |

**Table 3.** Details of the test specimens.

| Group ID | Cement (kg/m$^3$) | Sand (kg/m$^3$) | Stone (kg/m$^3$) | | |
|---|---|---|---|---|---|
| | | | 5–10 mm | 10–20 mm | 20–40 mm |
| 1 | 20 | 477 | 340.8 | 596.4 | 715.7 |
| 2 | 40 | 477 | 340.8 | 596.4 | 715.7 |
| 3 | 60 | 477 | 340.8 | 596.4 | 715.7 |
| 4 | 80 | 477 | 340.8 | 596.4 | 715.7 |
| 5 | 100 | 477 | 340.8 | 596.4 | 715.7 |

The preparation process of the specimens and experimental process of CSG material are the same as those of Yang et al., which is not described in detail here [14]. After each experiment, stress and strain were calculated according to collected data for analysis; these are shown in Figures 1–4.

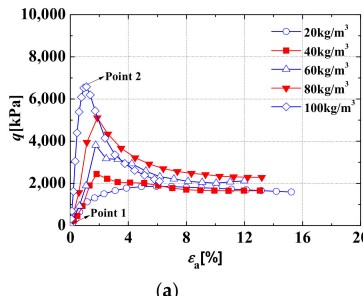
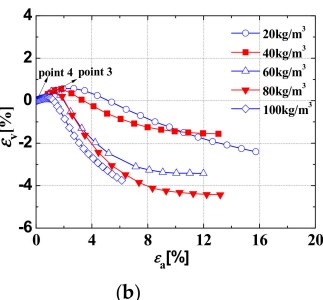

(a)  (b)

**Figure 1.** The curves of the triaxial tests for cemented sand and gravel (CSG) material with the confining pressure of 300 kPa: (**a**) the stress–strain curves; (**b**) the volumetric strain-axial strain curves.

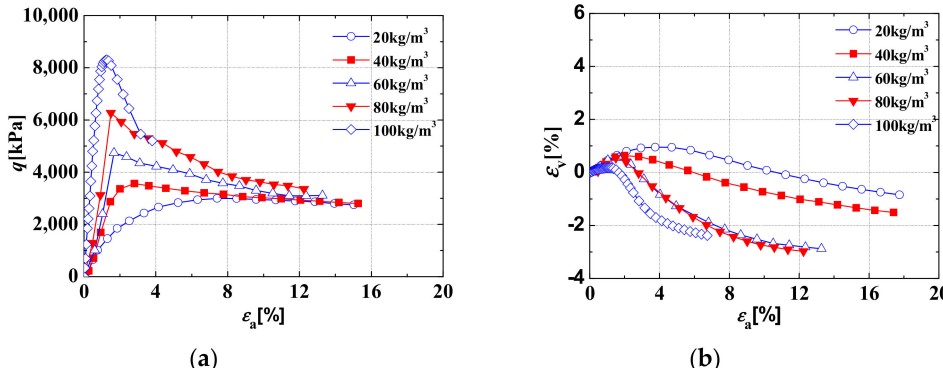

**Figure 2.** The curves of the triaxial tests for CSG material with the confining pressure of 600 kPa: (**a**) the stress–strain curves; (**b**) the volumetric strain-axial strain curves.

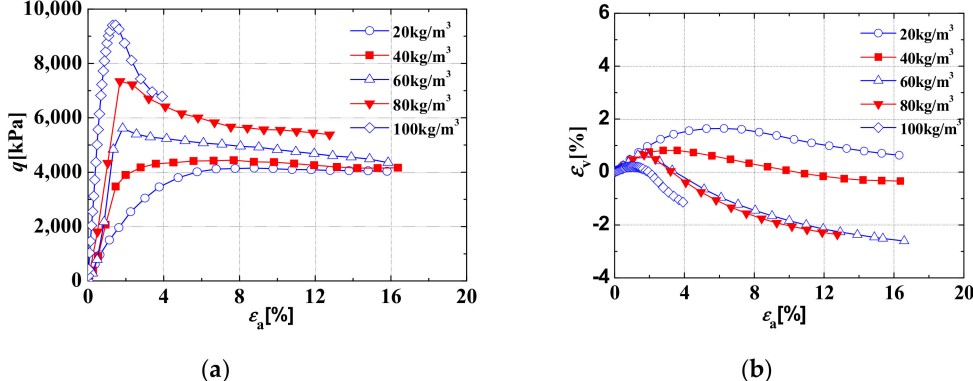

**Figure 3.** The curves of the triaxial tests for CSG material with the confining pressure of 900 kPa: (**a**) the stress–strain curves; (**b**) the volumetric strain-axial strain curves.

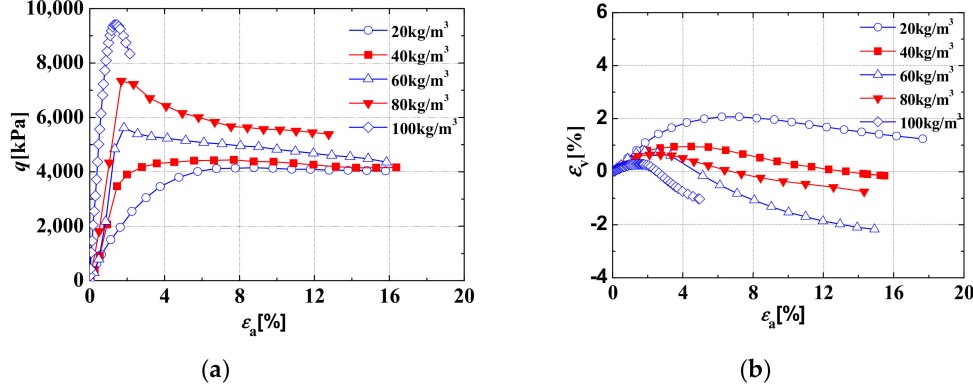

**Figure 4.** The curves of the triaxial tests for CSG material with the confining pressure of 1200 kPa: (**a**) the stress–strain curves; (**b**) the volumetric strain-axial strain curves.

## 3. Results and Discussion

### 3.1. Stress–Strain and Volumetric Strain–Axial Strain Curves

The stress–strain and strength characteristics in Figures 1a, 2a, 3a and 4a have been introduced by Yang et al., and these sections will not be discussed here [14]. The slope of the stress–strain curve when the axial strain is 0, called the initial modulus herein, and the strain at point 2 (the peak point of the stress–strain curve), called the failure strain, are key issues in the deformation calculation of CSG dams. Therefore, the effects of various cement contents and confining pressures on the initial modulus and failure strain are studied in this study.

It can be seen from Figures [1]b, [2]b, [3]b and [4]b that the volumetric strain–axial strain curves of the CSG material before the peak volumetric strain show obvious nonlinearity. When the cement content is low, the shear shrinkage and dilatancy of the CSG material are similar to those of the rockfill material; when the cement content is high, the dilatancy is more obvious, and the initial slopes of the volumetric strain–axial strain curves are less affected by the confining pressure. When the cement content is approximately 100 kg/m$^3$, the confining pressure has little influence on the initial slope of the CSG material. The volumetric strain and axial strain at point 3 (the peak point of the volumetric strain–axial strain curve) decrease with an increase in the cement content. The volumetric strain–axial strain characteristics of the CSG material above are roughly the same as those of polymer rockfill materials, sand reinforced with fibers, etc. [16,23].

As the CSG material begins to exhibit dilatancy at point 3, point 3 is also called the initial dilatancy point. The slope of the volumetric strain–axial strain curve when the axial strain is 0 (point 4) is called the initial tangent volumetric ratio here. These are also key quantities that are considered in the deformation calculation of CSG dams and quantitatively described below.

*3.2. Initial Modulus*

Figure [5] illustrates the initial modulus under varying confining pressures and cement contents. As shown in the figure, the initial modulus increases exponentially with an increase in confining pressure under a certain cement content. This is consistent with the observations made by previous studies on other types of CSG materials [10–12]. The relationship between the initial modulus and confining pressure of CSG material adopted by some of these studies is as follows [10]:

$$E_i = E_{01} Pa (\sigma_3 / Pa)^n, \tag{1}$$

where $E_{01}$ and $n$ are dimensionless parameters related to the type of the soil, rockfill material, etc.; the atmospheric pressure $Pa$ is 100 kPa; $E_{01} Pa$ represents the initial modulus when the confining pressure is 100 Pa; and $n$ represents the growth index of the initial modulus. Through a regression analysis of data from Figure [5] and Equation (1), the values of $E_{01}$ and $n$ are obtained and are shown in Table [4]. Values of Correlation coefficient $R^2$ in the table that are greater than 0.97 indicate that the calculated results of Equation (1) fit the experimental results well. However, when the confining pressure is 0, the initial modulus is 0 in Equation (1), and this is inconsistent with the actual value of the modulus. Thus, the relationship between the initial modulus and confining pressure for a CSG material, determined by Cai et al. and Fu et al., is expressed as follows [11,13]:

$$E_i = E_0 Pa (1 + \sigma_3 / Pa)^n, \tag{2}$$

where $E_0$ and $n$ are dimensionless parameters; $E_0 Pa$ represents the initial modulus of the CSG material when the confining pressure is 0; $n$ represents the growth index of the initial modulus. Through a regression analysis of data from Figure [6] and Equation (2), the values of $E_0$ and $n$ are obtained and are shown in Table [5]. Values of $R^2$ in the table that are greater than 0.98 indicate that the calculated results of Equation (2) fit the experimental results well, thereby demonstrating that Equation (2) can be used to describe the initial modulus of CSG material as a function of the confining pressure.

**Table 4.** Values of $R^2$, $E_{01}$, and $n$.

| Cc (kg/m$^3$) | Formula | Correlation Coefficient $R^2$ | $E_{01}$ | $n$ |
|---|---|---|---|---|
| 20 | $E_i = 86{,}078(\sigma_3/100)^{0.51}$ | 0.98 | 860.78 | 0.51 |
| 40 | $E_i = 109{,}766(\sigma_3/100)^{0.43}$ | 0.99 | 1097.66 | 0.43 |
| 60 | $E_i = 213{,}587(\sigma_3/100)^{0.35}$ | 0.99 | 2135.87 | 0.35 |
| 80 | $E_i = 366{,}396(\sigma_3/100)^{0.29}$ | 0.99 | 3663.96 | 0.29 |
| 100 | $E_i = 833{,}524(\sigma_3/100)^{0.16}$ | 0.97 | 8335.24 | 0.16 |

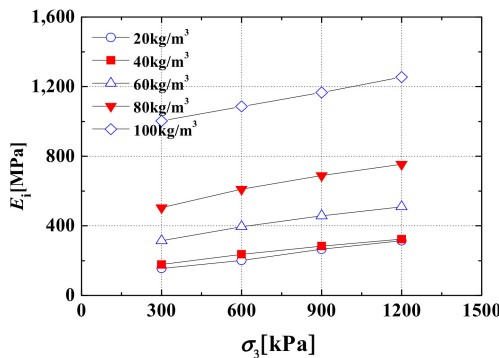

**Figure 5.** The relationship between the initial modulus $E_i$ and confining pressure $\sigma_3$ under varying cement contents.

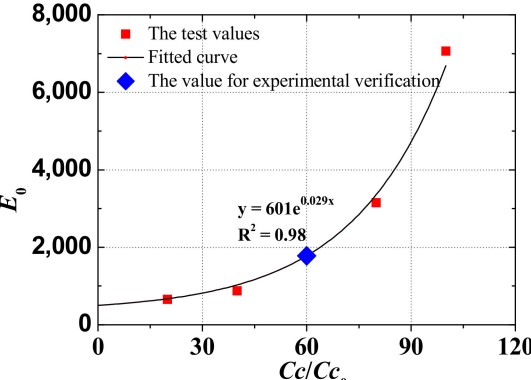

**Figure 6.** The relationship between the parameter $E_0$ and cement content $Cc$.

**Table 5.** Values of $R^2$, $E_0$, and $n$.

| Cc (kg/m$^3$) | Formula | Correlation Coefficient $R^2$ | $E_0$ | $n$ |
| --- | --- | --- | --- | --- |
| 20 | $E_i = 65,651(1 + \sigma_3/100)^{0.60}$ | 0.99 | 656.51 | 0.60 |
| 40 | $E_i = 109,766(1 + \sigma_3/100)^{0.45}$ | 0.99 | 1097.66 | 0.45 |
| 60 | $E_i = 178,000(1 + \sigma_3/100)^{0.41}$ | 0.99 | 1780.00 | 0.41 |
| 80 | $E_i = 315,000(1 + \sigma_3/100)^{0.31}$ | 0.99 | 3150.00 | 0.31 |
| 100 | $E_i = 706,177(1 + \sigma_3/100)^{0.20}$ | 0.98 | 7061.77 | 0.20 |

The test values of $E_0$ and $n$ under different cement contents are also shown in Figures 6 and 7, respectively. As can be seen from Figure 6, when $Cc$ is close to 0, $E_0$ is close to the corresponding value of rockfill material. With an increase of $Cc$, $E_0$ increases. The parameter $n$ is close to the corresponding value of rockfill material in Figure 7, when $Cc$ is close to 0. Additionally, with an increase of $Cc$, $n$ decreases and is close to 0.

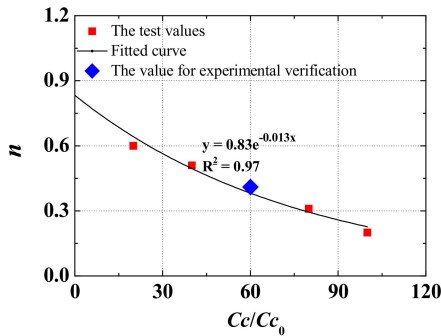

**Figure 7.** The relationship between the parameter $n$ and cement content $Cc$.

In the study, $E_0$ and $n$ were fitted with the corresponding parameters of CSG material for cement contents of 20, 40, 80, and 100 kg/m$^3$; we thus get

$$E_0 = E'_0 e^{b'(Cc/Cc_0)},\qquad(3)$$

$$n = n'_0 e^{c'(Cc/Cc_0)},\qquad(4)$$

where $E'_0$ and $b'$ are the fitting coefficients of the relationship between $E_0$ and $Cc$; $n'_0$ and $c'$ are the fitting coefficients of the relationship between $n$ and $Cc$; $Cc_0$ is reference cement content with a value of 1 kg/m$^3$. The curves fitted by Equations (3) and (4) are shown in Figures 6 and 7, respectively. Values of $R^2$ in these curves that are greater than 0.97 indicate that the calculated results of Equations (3) and (4) can fit the experimental results well.

From Equations (2)–(4), we can derive the relationship of the initial modulus of CSG material with the cement content and confining pressure as follows:

$$E_i = E'_0 e^{b'(Cc/Cc_0)} Pa(1 + \sigma_3/Pa)^{n'_0 e^{c'(Cc/Cc_0)}}.\qquad(5)$$

Equation (5) can quantitatively reflect the influence of confining pressure and cement content on the initial modulus $E_i$: when the cement content $Cc$ is 0, this equation can be used to describe the relationship between the initial modulus of the rockfill material and confining pressure; with an increase in the cement content, $E_i$ under a certain confining pressure increases; when the cement content increases to 80 kg/m$^3$, $E_i$ is less affected by the confining pressure.

### 3.3. Axial Strain at the Peak Points of the Test Curves

In this study, the axial strain at the peak points of the stress–strain curves represent the failure strain, and the axial strain at the peak points of the volumetric strain–axial strain curves represents the axial strain at the initial dilatancy point. Figure 8 shows that these axial strains decrease with increasing cement content and decreasing confining pressure. However, the value of failure strain is larger than that of the axial strain at the initial dilatancy point under the same cement content and confining pressure. This suggests that the dilatancy is usually observed before failure of the CSG material. When the cement content is higher than 60 kg/m$^3$, the failure strain is almost the same and between 1.2% and1.5% under a certain confining pressure, which indicates that the brittleness of CSG material is more obvious with increasing cement content. The axial strain at the initial dilatancy point of CSG material with a certain cement content is also almost the same and between 0.9% and1.3%. According to the analysis in Figure 8, the relationship between failure strain $\varepsilon_m$ and confining pressure and the relationship between axial strain at the initial dilatancy point $\varepsilon_n$ and confining pressure can be expressed as follows:

$$\varepsilon_m = \lambda_0(\sigma_3/Pa) + d_0,\qquad(6)$$

$$\varepsilon_n = \lambda_1(\sigma_3/Pa) + d_1,\qquad(7)$$

where $\lambda_0$ and $\lambda_1$ are dimensionless parameters related to the cement content and the slope of those lines; $d_0$ and $d_1$ are the vertical intercept and dimensionless parameters related to the cement content. Through a regression analysis of data from Figure 8, the values of $E_{01}$ and $n$ in Equation (6) are shown in Table 6. The values of $E_0$ and $n$ in Equation (7) can be obtained and are shown in Table 7. Values of $R^2$ in those tables that are greater than 0.95 indicate that the calculated results of Equations (6) and (7) can fit the experimental results well.

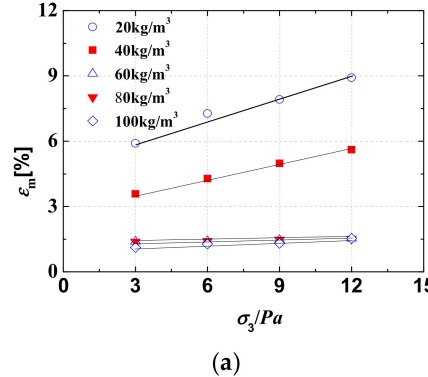 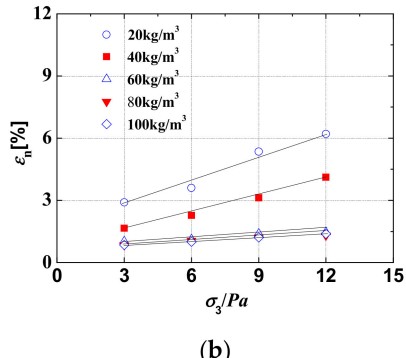

(**a**)     (**b**)

**Figure 8.** The relationship between the axial strain and confining pressure $\sigma_3$: (**a**) the failure strain $\varepsilon_m$; (**b**) the axial strain at the initial dilatancy point $\varepsilon_n$.

**Table 6.** Values of $R^2$, $\lambda_0$, and $d_0$.

| $Cc$ (kg/m$^3$). | Formula | Correlation Coefficient $R^2$ | $\lambda_0$ (%) | $d_0$ (%) |
|---|---|---|---|---|
| 20 | $\varepsilon_m = 0.33(\sigma_3/100) + 4.7$ | 0.98 | 0.33 | 4.7 |
| 40 | $\varepsilon_m = 0.20(\sigma_3/100) + 1.92$ | 0.99 | 0.2 | 1.92 |
| 60 | $\varepsilon_m = 0.065(\sigma_3/100) + 1.22$ | 0.96 | 0.065 | 1.22 |
| 80 | $\varepsilon_m = 0.058(\sigma_3/100) + 0.99$ | 0.98 | 0.058 | 0.99 |
| 100 | $\varepsilon_m = 0.041(\sigma_3/100) + 0.96$ | 0.97 | 0.041 | 0.96 |

**Table 7.** Values of $R^2$, $\lambda_1$, and $d_1$.

| $Cc$ (kg/m$^3$) | Formula | Correlation Coefficient $R^2$ | $\lambda_1$(%) | $d_1$(%) |
|---|---|---|---|---|
| 20 | $\varepsilon_n = 0.36(\sigma_3/100) + 2.29$ | 0.97 | 0.36 | 2.29 |
| 40 | $\varepsilon_n = 0.23(\sigma_3/100) + 1.23$ | 0.99 | 0.23 | 1.23 |
| 60 | $\varepsilon_n = 0.082(\sigma_3/100) + 0.84$ | 0.95 | 0.082 | 0.84 |
| 80 | $\varepsilon_n = 0.053(\sigma_3/100) + 0.72$ | 0.96 | 0.053 | 0.72 |
| 100 | $\varepsilon_n = 0.041(\sigma_3/100) + 0.65$ | 0.97 | 0.041 | 0.65 |

By fitting the test values of $\lambda_0$ and $d_0$ for different cement contents, which are shown in Figures 9 and 10 of CSG material with the cement contents of 20, 40, 80, and 100 kg/m$^3$, we have

$$\lambda_0 = a_0 e^{-c_0(Cc/Cc_0)}, \tag{8}$$

$$d_0 = l_0 e^{-m_0(Cc/Cc_0)} + n_0, \tag{9}$$

where $a_0$ and $c_0$ are the fitting coefficients of the relationship between $\lambda_0$ and $Cc$; $l_0$, $m_0$, and $n_0$ are the fitting coefficients of the relationship between $d_0$ and $Cc$. The curves fitted by Equations (8) and (9) are shown in Figures 9 and 10, respectively. Values of $R^2$ that are greater than 0.99 indicate that the calculated results of Equations (8) and (9) fit the experimental results well.

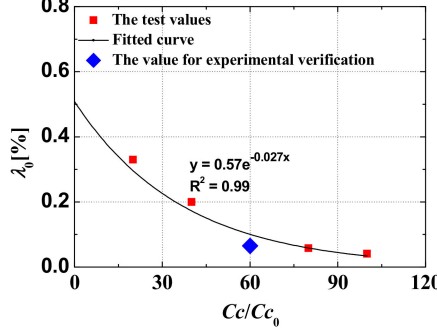

**Figure 9.** The relationship between the parameter $\lambda_0$ and cement content $Cc$.

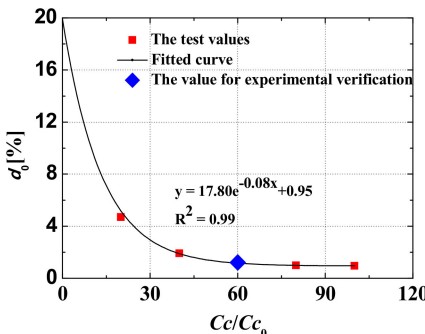

**Figure 10.** The relationship of the parameter $d_0$ and cement content $Cc$.

The test values of $\lambda_1$ and $d_1$ under different cement contents are shown in Figures 11 and 12, respectively. By fitting the data of the triaxial shear test with the cement contents of 20, 40, 80, and 100 kg/m$^3$, we have

$$\lambda_1 = a_1 e^{-c_1(Cc/Cc_0)}, \tag{10}$$

$$d_1 = l_1 e^{-m_1(Cc/Cc_0)} + n_1, \tag{11}$$

where $a_1$ and $c_1$ are the fitting coefficients of the relationship between $\lambda_1$ and $Cc$; $l_1$, $m_1$, and $n_1$ are the fitting coefficients of the relationship between $d_1$ and $Cc$. The curves fitted by Equations (10) and (11) are shown in Figures 11 and 12, respectively. Values of R$^2$ that are greater than 0.98 indicate that the calculated results of Equations (10) and (11) fit the experimental results well.

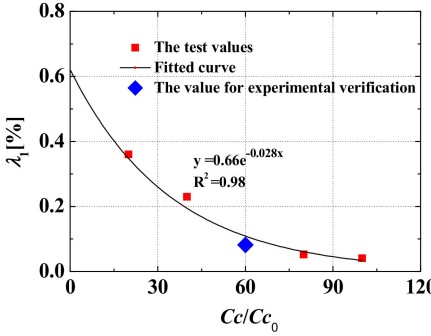

**Figure 11.** The relationship between parameter $\lambda_1$ and cement content $Cc$.

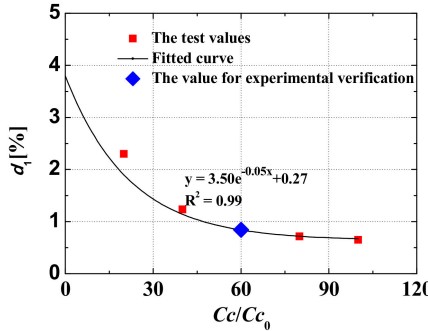

**Figure 12.** The relationship of parameter $d_1$ and cement content $Cc$.

By substituting Equations (8) and (9) into Equation (6) and Equations (10) and (11) into Equation (7), we can derive the relationship between failure strain $\varepsilon_\mathrm{m}$ and cement content $Cc$ and confining pressure, and the relationship between the axial strain at the initial dilatancy point $\varepsilon_\mathrm{n}$ and the cement content $Cc$ and confining pressure as follows:

$$\varepsilon_{\mathrm{m}} = a_0 e^{-c_0(Cc/Cc_0)}(\sigma_3/Pa) + l_0 e^{-m_0(Cc/Cc_0)} + n_0, \tag{12}$$

$$\varepsilon_{\mathrm{n}} = a_1 e^{-c_1(Cc/Cc_0)}(\sigma_3/Pa) + l_1 e^{-m_1(Cc/Cc_0)} + n_1. \tag{13}$$

### 3.4. Volumetric Strain at the Initial Dilatancy Point

The volumetric strain at the initial dilatancy point is referred to as the peak volumetric strain $\varepsilon_{\mathrm{vd}}$ in this study and decreases with increasing cement content and decreasing confining pressure, as shown in Figure 13. It can be expressed as

$$\varepsilon_{\mathrm{vd}} = \lambda_2(\sigma_3/Pa) + d_2, \tag{14}$$

where $\lambda_2$ is a dimensionless parameter related to the cement content and the slope of lines; $d_2$ is a dimensionless parameter related to the cement content and the vertical intercept. The curves fitted by Equation (14) are shown in Table 8. Values of $R^2$ that are greater than 0.96 indicate that the calculated results of Equation (14) fit the experimental results well.

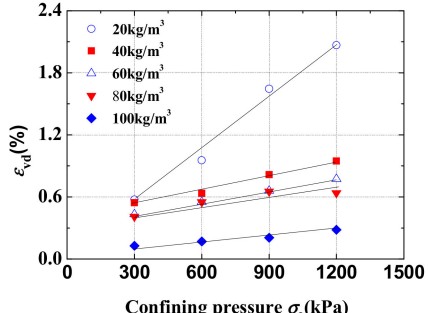

**Figure 13.** The relationship between the peak volumetric strain $\varepsilon_{\mathrm{vd}}$ and confining pressure $\sigma_3$.

**Table 8.** Values of $R^2$, $\lambda_2$, and $d_2$.

| Cc (kg/m$^3$) | Formula | Correlation Coefficient $R^2$ | $\lambda_2$ (%) | $d_2$ (%) |
|---|---|---|---|---|
| 20 | $\varepsilon_{\mathrm{vd}} = 0.96(\sigma_3/100) + 0.50$ | 0.99 | 0.96 | 0.50 |
| 40 | $\varepsilon_{\mathrm{vd}} = 0.44(\sigma_3/100) + 0.37$ | 0.99 | 0.44 | 0.37 |
| 60 | $\varepsilon_{\mathrm{vd}} = 0.29(\sigma_3/100) + 0.26$ | 0.99 | 0.29 | 0.26 |
| 80 | $\varepsilon_{\mathrm{vd}} = 0.24(\sigma_3/100) + 0.17$ | 0.96 | 0.24 | 0.17 |
| 100 | $\varepsilon_{\mathrm{vd}} = 0.17(\sigma_3/100) + 0.10$ | 0.97 | 0.17 | 0.10 |

The peak volumetric strain $\varepsilon_{\mathrm{vd}}$ is also affected by the cement content, and Figures 14 and 15 shows the test values of $\lambda_2$ and $d_2$ under different cement contents with the cement contents of 20, 40, 80, and 100 kg/m$^3$. As can be seen from Figure 14, with increasing $Cc$, $\lambda_2$ gradually decreases. When $Cc$ tends to the cement content of the roller compacted concrete, $\lambda_2$ is close to 0, which indicates that the compaction of the CSG material increases and the shear shrinkage decreases with the increasing cement content. The parameter $\lambda_2$ can be expressed as

$$\lambda_2 = a_2 e^{-c_2(Cc/Cc_0)}, \tag{15}$$

where $a_2$ and $c_2$ are the fitting coefficients of the relationship between $\lambda_2$ and $Cc$.

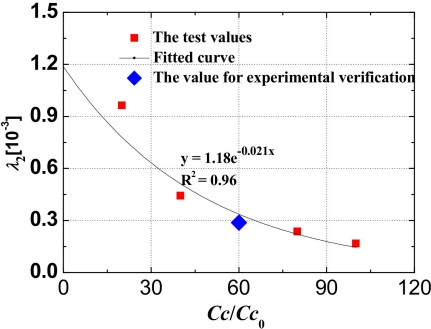

**Figure 14.** The relationship between the parameter $\lambda_2$ and cement content *Cc*.

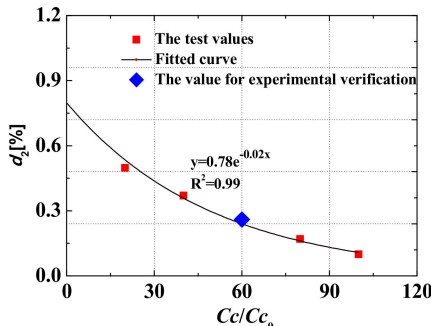

**Figure 15.** The relationship between the parameter $d_2$ and cement content *Cc*.

As can be seen from Figure 15, the parameter $d_2$ decreases with the increase of the cement content, which can be expressed as follows:

$$d_2 = l_2 e^{-m_2(Cc/Cc_0)}, \tag{16}$$

where $l_2$ and $m_2$ are the fitting coefficients of the relationship between $d_2$ and *Cc*.

The curves fitted by Equations (15) and (16) are shown in Figures 14 and 15, respectively. Values of $R^2$ that are greater than 0.96 indicate that the calculated results of Equations (15) and (16) fit the experimental results well. When *Cc* is close to 0, $d_2$ in Equation (16) is close to $l_2$, the peak volumetric strain increases, and CSG material presents an obvious shear shrinkage phenomenon. When *Cc* increases, $d_2$ gradually tends to 0, and CSG material firstly exhibits a small shear shrinkage followed by obvious dilatancy.

By substituting Equations (15) and (16) into Equation (14), the peak volumetric strain of CSG material with a specific cement content and confining pressure is expressed as:

$$\varepsilon_{vd} = a_2 e^{-c_2(Cc/Cc_0)}(\sigma_3/Pa) + l_2 e^{-m_2(Cc/Cc_0)}. \tag{17}$$

### 3.5. Initial Tangent Volumetric Ratio

Liu et al. directly assumed that there was a linear relationship between volumetric strain and axial strain in the nonlinear elastic model of CSG material [24]. However, the volumetric strain and axial strain of CSG material had obvious nonlinear characteristics when the cement contents were low, so the feature of the curves was difficult to represent as a straight line. The cubic polynomial adopted by Cai et al. can represent the relationship of the volumetric strain and axial strain well, but this relationship is more complicated [25]. The volumetric-strain–axial-strain relationship of the rockfill material that can adequately reflect the dilatancy characteristics of CSG material is expressed by a quadratic function that passes through the origin (0,0) as follows:

$$\varepsilon_v = A\varepsilon_a^2 + B\varepsilon_a, \tag{18}$$

where $A$ and $B$ are the material parameters.

To reflect the physical significance of parameters in the volumetric-strain–axial-strain relationship of CSG material, Equation (18) can be rewritten as

$$\varepsilon_v = A(\varepsilon_a - \varepsilon_n)^2 + \varepsilon_{vd}, \tag{19}$$

where $\varepsilon_n$ is the axial strain of the peak point, which can be obtained from Equation (19), $d$ is the fitting coefficient, and $\varepsilon_{vd}$ is the peak volumetric strain.

Equation (19) can reasonably reflect the shear shrinkage and dilatancy characteristics of CSG material. The tangent volumetric ratio $\mu_t$ is derived as follows:

$$\mu_t = \frac{d\varepsilon_v}{d\varepsilon_a} = 2A(\varepsilon_a - \varepsilon_n). \tag{20}$$

As the origin (0, 0) in the volumetric-strain–axial-strain curve is drawn according to the triaxial shear test results, the point is substituted into Equation (19), and $A$ can be expressed as follows:

$$A = -\varepsilon_{vd}/\varepsilon_n{}^2. \tag{21}$$

When the axial strain $\varepsilon_a$ is 0, $\mu_t$ is the initial tangent volumetric ratio is as follows:

$$\mu_t = 2\varepsilon_{vd}/\varepsilon_n. \tag{22}$$

According to Equation (19), the initial tangent volumetric ratio $\mu_t$ for CSG material is related to the peak volumetric strain and its corresponding axial strain.

*3.6. Rationality of Expressions for the Deformation Properties of CSG Material*

Equations (5), (12), (13), (17), and (22) are the expressions for quantified indexes of the deformation characteristics of CSG material. To verify the rationality of these expressions, the experimental results of CSG material with the cement content of 60 kg/m$^3$ for the experimental verification, and the calculated results are shown in Figures 6, 7, 9–12, 14 and 15. The calculated results fit the experimental results well, thereby demonstrating that these equations can be used to describe the deformation properties of CSG material under different cement contents.

To verify the rationality of these expressions determining the deformation properties in different types of CSG materials and some other cemented granular materials, Figures 16 and 17 show the test results of CSG materials parameters obtained by Yang and Fu et al. (the maximum size of the gravel grains is 40 mm or greater), the test results of cemented coarse-gained soil parameters (the maximum size of the gravel grains is 20 mm), and cemented poorly graded sand–gravel mixture parameters (the maximum size of the gravel grains is 12.5 mm) [13,15,26,27]. The type, chemical and physical properties of the raw materials included cement and coarse and fine aggregates of those are different from those used in this study.

The fitting functions, i.e., Equations (3), (4), (8)–(11), (15) and (16) for the parameters in Equations (5), (12), (13), (17), and (22) under different cement contents can be used to fit the test results of CSG materials parameters by Yang and Fu et al. as well as the test results of cemented coarse-gained soil parameters by Li in Figure 16 [13,26,27]. These verification results demonstrate that the expressions for quantified indexes of the deformation characteristics can also be used to well describe the deformation characteristics of other types of CSG materials and cemented coarse-gained soil.

The expressions for the deformation properties of CSG material are used to simulate the deformation characteristics of a cemented poorly graded sand–gravel mixture as shown in Figure 17 [15]. However, some expressions considering the effect of confining pressure cannot fit the corresponding test results for the material well. The main reason could be that the grading and size of the aggregate

particles, cement content, and confining pressure of the cemented poorly graded sand–gravel mixture are different with those of the cemented granular material described above.

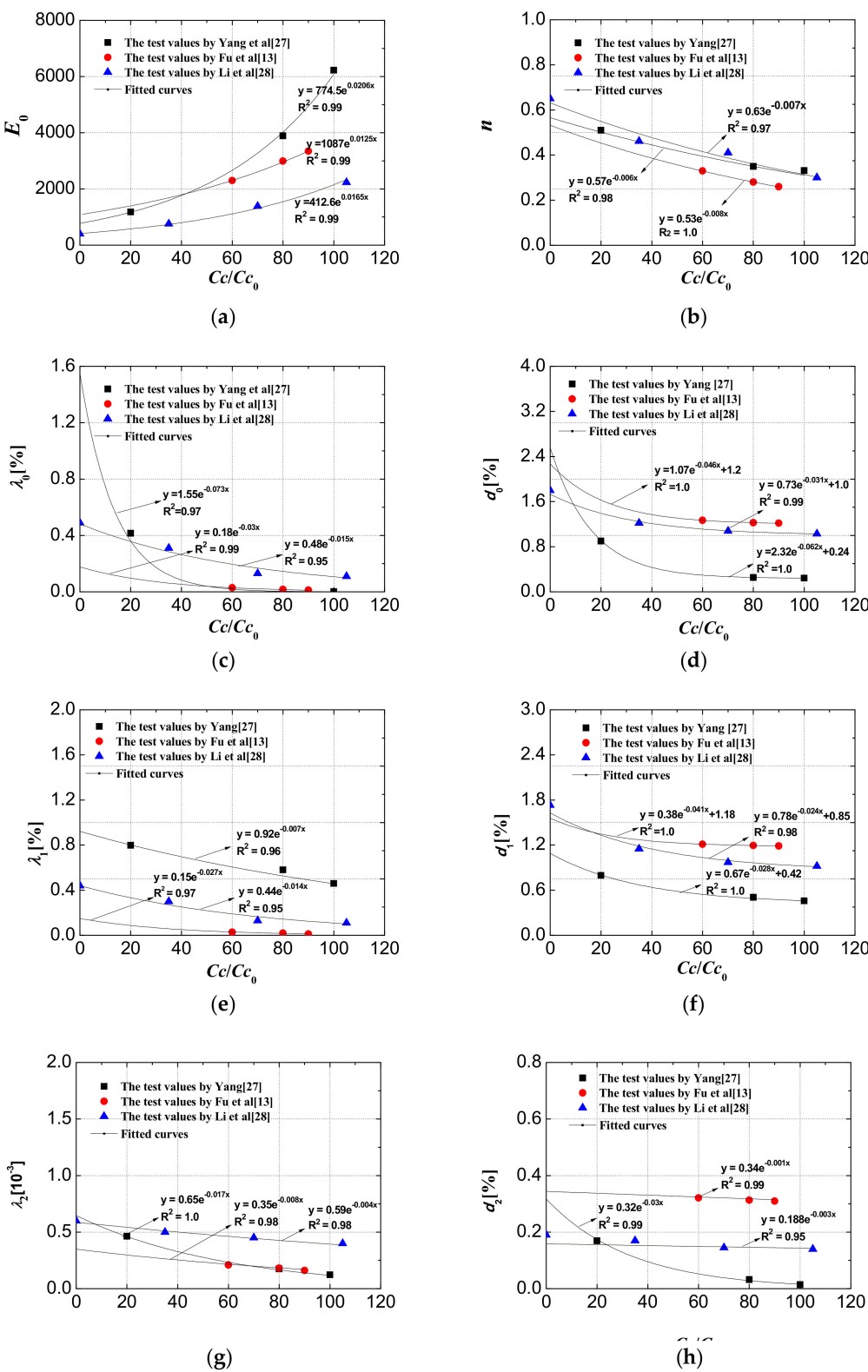

**Figure 16.** The relationship between the parameters and the cement content *Cc*: (**a**) $E_0$; (**b**) $n$; (**c**) $\lambda_0$; (**d**) $d_0$; (**e**) $\lambda_1$; (**f**) $d_1$; (**g**) $\lambda_2$; (**h**) $d_2$.

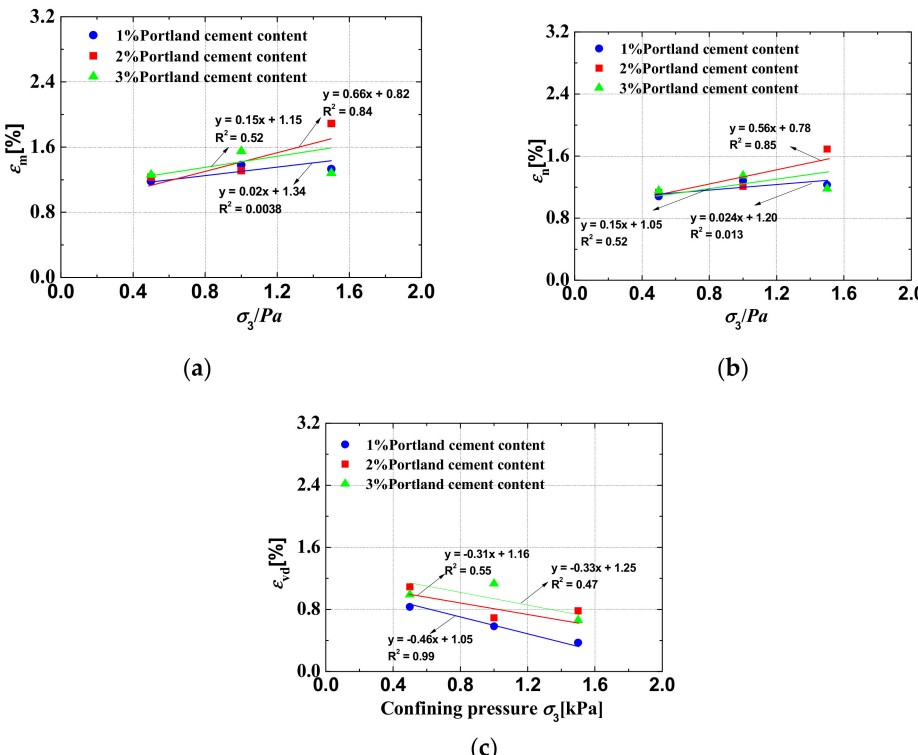

**Figure 17.** The verification of the expressions for the deformation characteristics of the cemented poorly graded sand–gravel mixture under different cement contents: (**a**) the relationship between the failure strain $\varepsilon_m$ and confining pressure $\sigma_3$; (**b**) the relationship between the axial strain of the peak point $\varepsilon_n$ and confining pressure $\sigma_3$; (**c**) the relationship between the peak volumetric strain $\varepsilon_{vd}$ and confining pressure $\sigma_3$.

In particular, these expressions including Equations (5), (12), (13), (17), and (22) in this study can be considered to be applicable to the cemented granular materials with a cement content less than 100 kg/m$^3$ and a maximum particle size of the aggregate greater than 20 mm under different confining pressures.

*3.7. Mechanism Analysis*

To further understand the effect of cement content on the deformation characteristics of CSG material, the internal mechanism of the material under a triaxial load is analyzed here.

When the cement content is low, there is large porosity in the CSG material and weak cementation between aggregates. In the initial shearing stage of the CSG material, the interaction of the particles is mainly friction and extrusion, which is similar to that of ordinary rockfill material; thus, the initial modulus of the CSG material is small and close to that of the rockfill material, and the shear shrinkage is shown. Then, axial load makes the internal porosity decrease, and the particles are mainly flipped and spanned, which can cause dilatancy of the CSG material. After that, the particles near the shear plane are staggered after overcoming friction, and weak cementation is produced by cement in the CSG material; the CSG material is destructed and its failure strain is approximately 6%–10%. As the axial load continues to increase, the particles near the shear plane continue to overcome the friction, the frictional resistance makes the residual strength of the CSG material slightly less than the failure strength, and the stress–strain curve is similar to that of the ordinary rockfill [28].

With an increase of cement content, the internal cementation of the CSG material increases and porosity decreases. The initial modulus increases as shown in Figure 5, and the shear shrinkage becomes less obvious as shown in Figures 1b, 2b, 3b and 4b. When some particles of cementation are broken, then flipped and spanned, dilatancy occurs in the CSG material; the axial strain and volumetric

strain of the initial dilatancy decreases gradually with increasing cement content, as shown in Figures 8b and 13. When particles near the shear surface continue to be loaded, the cementation is broken, but the smaller dislocation results in the smaller failure strain as shown in Figure 11 when the cement content is higher. In addition, since there are many loose particles in the shear zone after the failure of the specimen, and there is randomness in its crossing and turning, the volumetric-strain–axial-strain curve of the CSG material with different cement content presents an occasional crossover phenomenon, as shown in Figures 1b, 2b, 3b and 4b.

## 4. Conclusions

In this study, a series of triaxial compressive experiments were carried out on CSG materials for cement contents of 20, 40, 60, 80, and 100 kg/m$^3$ with a confining pressure range of 0.3–1.2 MPa. Based on the experimental results, the stress–strain behavior, initial modulus, and the initial Poisson's ratio of CSG material were analyzed. The conclusions obtained are as follows:

(1)  The predictive model for the failure strain considering the effects of the confining pressure and the cement content is established. The model shows that the failure strain increases linearly with increasing confining pressure under a certain cement content; the failure strain decreased, and the brittleness of CSG material increased with increasing cement content.

(2)  The expression for initial modulus of CSG material considering the influence of cement content and confining pressure was proposed. This revealed that the modulus of the CSG material increases exponentially with increasing cement content.

(3)  Expressions for peak volumetric strain and its corresponding axial strain with the amount of cement content and confining pressure were proposed. These revealed that the peak volumetric strain and its corresponding axial strain increase linearly with increasing confining pressure, and the peak volumetric strain and its corresponding axial strain decrease with the increase in cement content.

(4)  The predictive model for the initial tangent volumetric ratio is established, and this value can be determined by the peak volumetric strain and its corresponding axial strain.

(5)  The expressions in this study are applicable when determining deformation properties of cemented granular materials with a cement content less than 100 kg/m$^3$ and a maximum particle size of the aggregates greater than 20 mm under different confining pressures and cement contents.

These conclusions can help construct a reasonable constitutive model for the various CSG materials under different cement contents and can promote the application and utilization of CSG materials in dams, embankments, roadbeds, and so on.

**Author Contributions:** Conceptualization, X.C.; data curation, J.Y.; funding acquisition, X.C.; investigation, J.Y. and X.G.; validation, X.C. and X.G.; writing—original draft, J.Y. and J.Z.; writing—review & editing, J.Z.

**Funding:** This research was funded by the National Key Research and Development Program of China (2018YFC0406804), and the National Natural Science Foundation of China (51179061).

**Conflicts of Interest:** The authors declare no conflict of interest.

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
