# Peer review of "Effect of Cement Content on the Deformation Properties of Cemented Sand and Gravel Material"

_applsci, doi:10.3390/app9112369_

Round 1

Reviewer 1 Report

This paper needs major revisions as follows: 1. Abstract should be revised. It should include the aim of the study, study methodology and specific results. 2. Introduction needs to be updated, there are several related studies available which authors have missed. 3. Used raw materials’ chemical/physical/mechanical properties must be added to the paper. 4. For the ‘Results and discussion’ section, more available data from literature can be added to develop more reliable equations. 5. Conclusions section can be improved with more specific findings.

Author Response

Point 1: Abstract should be revised. It should include the aim of the study, study methodology and specific results.

Response 1: The Abstract has been revised.

Point 2: Introduction needs to be updated, there are several related studies available which authors have missed.

Response 2: Several related studies has been added to the Introduction

Point 3: Used raw materials’ chemical/physical/mechanical properties must be added to the paper.

Response 3: The used raw materials’ physical properties have been added in lines 98-104 of the revised paper.

Point 4: For the ‘Results and discussion’ section, more available data from literature can be added to develop more reliable equations.

Response 4: The section has been modified.

Point 5:Conclusions section can be improved with more specific findings.

Response 5: The section has been modified.

Reviewer 2 Report

The authors report the study of the effect of different cement contents on the deformation properties of cemented materials. Starting from some triaxial shear stress tests, they discuss the relationship between many different parameters and the cement content, and they gained information on different deformation characteristics, such as the initial modulus, axial strain and peak volumetric strain.

The authors carried out a series of triaxial compressive experiments when adding 20, 40, 60, 80 or 100 kg/m3 of cement to sand and gravel materials, with a confining pressure of 0.3, 0.6, 0.9 and 1.2 MPa. The analysis of the data allowed the authors to reveal that, in this kind of materials, the modulus increases when increasing the cement content. On the opposite, the peak volumetric strain decreases when increasing the cement content, and increases when increasing the confining pressure.

Although the analysis of the data here presented is interesting, the results are not clearly discussed and the novelty of this article with respect to the previous studies should be better introduced. The specimens and the compressive experiments here shown were already presented by some of the authors in a previous paper (ref 14). This should be clearly stated. Figures 1a, 2a, 3a and 4a seem to correspond to Figure 3 of reference 14. Since the samples preparation and the test methods were already published somewhere else, this part should be only briefly described and appropriately cited in the Materials and Methods section.

Moreover, the equations used in the discussion should be better explained, also by defining all the parameters, and the proper references should be cited.

Some of the findings claimed by the authors could be predictable, but the overall analysis of the data lead to some interesting additional information with respect to the previous paper published by some of the authors. I suggest to the authors to heavily revise the whole work, both formally and substantially, to clearly explain the novelty of this article, to improve the discussion and to respect the author guidelines of the journal.

In addition to the major concerns detailed above, below are some specific comments:

·      Abbreviations should be defined the first time they appear. Please, define RCC (line 70).

·      Refer to the instruction for Authors to include the appropriate sections in your Manuscript.  

·      The Results and Discussion section is not clearly presented and it is hard to follow the discussion and to appreciate the importance of each figure showing the relationship between different parameters. I would also reduce the use of subsections.

·      Since the authors of ref. 16 have already investigated through triaxial compression test the effect of cement content on the properties of artificial cemented sand materials, what is the novelty of the the work presented in this paper? You need to consider this point.

·      In general, the explanations of the equations are not satisfactory and the parameters should be described more accurately.

·      The error bars are missing in all the Figures. When fitting some experimental data with models, it is particularly important to take into account the associated error.

·      The point 1 of the conclusions was already stated in ref 14 and it is not a direct result of the work presented in this article.

·      You need to check the style of the references. Ref 2, 3, 12, 20 and 21 were not correctly cited. 

Author Response

Point 1: Although the analysis of the data here presented is interesting, the results are not clearly discussed and the novelty of this article with respect to the previous studies should be better introduced. The specimens and the compressive experiments here shown were already presented by some of the authors in a previous paper (ref 14). This should be clearly stated. Figures 1a, 2a, 3a and 4a seem to correspond to Figure 3 of reference 14. Since the samples preparation and the test methods were already published somewhere else, this part should be only briefly described and appropriately cited in the Materials and Methods section.

Response 1: This section has been modified.

Point 2: Moreover, the equations used in the discussion should be better explained, also by defining all the parameters, and the proper references should be cited.

Response 2: This section has been modified.

Point 3: Some of the findings claimed by the authors could be predictable, but the overall analysis of the data lead to some interesting additional information with respect to the previous paper published by some of the authors. I suggest to the authors to heavily revise the whole work, both formally and substantially, to clearly explain the novelty of this article, to improve the discussion and to respect the author guidelines of the journal.

Response 3: The whole work of the paper has been revised. The novelty of this article is that some prediction models are established which can reveal the effect on the deformation characteristics of CSG material and provide a theoretical basis for improving the accuracy of the strain–stress forecasting results of CSG material for applications of the CSG dam and other projects. The discussion of this paper has been improved.

Point 4: Abbreviations should be defined the first time they appear. Please, define RCC (line 70).

Response 4: RCC is roller compacted concrete, and this problem has been modified in the revised paper.

Point 5:  Refer to the instruction for Authors to include the appropriate sections in your Manuscript.

Response 5: This problem has been modified.

Point 6:  The Results and Discussion section is not clearly presented and it is hard to follow the discussion and to appreciate the importance of each figure showing the relationship between different parameters. I would also reduce the use of subsections.

Response 6: The Results and Discussion section has been modified.

Point 7: Since the authors of ref. 16 have already investigated through triaxial compression test the effect of cement content on the properties of artificial cemented sand materials, what is the novelty of the the work presented in this paper? You need to consider this point.

Response 7: The authors of ref. 16 have already investigated through triaxial compression test the effect of cement content on the strength characteristics of artificial cemented sand materials. Howrver, the deformation properties of the artificial cemented sand materials have not been explored in ref. 16. In this study, some indexes of the deformation characteristics such as the failure strain, initial modulus, peak volumetric strains and the initial tangent volumetric ratio under different cement contents and confining pressures are quantified. And those indexes can be used to predicted the corresponding results under other cement contents and  confining pressures.

Point 8:   In general, the explanations of the equations are not satisfactory and the parameters should be described more accurately.

Response 8: This problem has been modified.

Point 9: The error bars are missing in all the Figures. When fitting some experimental data with models, it is particularly important to take into account the associated error.

Response 9: This problem has been modified.

Point 10: The point 1 of the conclusions was already stated in ref 14 and it is not a direct result of the work presented in this article.

Response 10: The stress–strain characteristics and the strength characteristics under different confining pressures and  cement contents was already stated in ref 14.However, the predictive model of the failure stain is firstly proposed in this paper. Point 11: You need to check the style of the references. Ref 2, 3, 12, 20 and 21 were not correctly cited.

Response 11: The problem has been modified.

Round 2

Reviewer 1 Report

1.       Following references should be considered in the introduction and analytical sections:

Mechanical properties of conventional and self-compacting concrete: an analytical study. Construction and Building Materials, 36, 330-347.

Self-compacting concrete incorporating steel and polypropylene fibers: compressive and tensile strengths, moduli of elasticity and rupture, compressive stress-strain curve, and energy dissipated under compression. Composites Part B: Engineering, 53, 121-133.

2.       This comment did not considered properly: Used raw materials’ chemical/physical/mechanical properties must be added to the paper. These properties should be presented as tables or graphs.

3.       This comment did not considered properly: For the ‘Results and discussion’ section, more available data from literature can be added to develop more reliable equations.

Author Response

Point 1:  Following references should be considered in the introduction and analytical sections:

Mechanical properties of conventional and self-compacting concrete: an analytical study. Construction and Building Materials, 36, 330-347.

Self-compacting concrete incorporating steel and polypropylene fibers: compressive and tensile strengths, moduli of elasticity and rupture, compressive stress-strain curve, and energy dissipated under compression. Composites Part B: Engineering, 53, 121-133.

Response 1: Those references have been considered in the introduction and analytical sections and are at lines 98-106 as follows: Farhad et al. conducted a comparative analysis between the conventional models in terms of evaluating mechanical properties of self-compacting and conventional concrete and developed new modulus of elasticity models, tensile strength models, and compressive stress–strain models for those materials; In addition, Farhad et al. investigated the compressive and splitting tensile strengths, modulus of elasticity and rupture, compressive stress–strain curve, and energy dissipated under compression at different curing ages for a control self-compacting concrete (SCC) mixture and three fiber-reinforced SCC containing steel, polypropylene, and hybrid (steel + polypropylene) fibers, and established their corresponding prediction models considering the effect of curing age.

Point 2: This comment did not considered properly: Used raw materials’ chemical/physical/mechanical properties must be added to the paper. These properties should be presented as tables or graphs.

Response 2: This section has been modified and is as follows:

Table 1. Physical properties and composition of crushed stones and sand

Aggregate type

Specific gravity

Bulk density

(kg/m3)

Water content

Clay content

Crushed stone

2.71

1650

0.01%

0.01%

Sand

2.62

1450

0.01%

0.01%

Table 2. Physical properties and chemical compositionof the cement

The fineness

The content of SO3

The content of MgO

2.26%

2.56%

1.78%

Point 3: This comment did not considered properly: For the ‘Results and discussion’ section, more available data from literature can be added to develop more reliable equations.

Response 3:

More available data from literatures by Yang and Li has been added to develop more reliable equations and is as follows:

Equations (5), (12), (13), (17), and (22) are the expressions for quantified indexes of the deformation characteristics of CSG material. To verify the rationality of these expressions, the experimental results of CSG material with the cement content of 60 kg/m3, and the calculated results are shown in Figures 6, 7, 9, 10, 11, 12, 14, and 15. The calculated results fit the experimental results well, thereby demonstrating that these equations can be used to describe the deformation properties of CSG material under different cement contents.

To verify the rationality of these expressions when determining the deformation properties of in different types of CSG materials and some other cemented granular materials, Figures 16 and 17 show the test results ofCSG materials  parameters obtained by  Yang and Fu et al.(The maximum size of the gravel grains is 40mm or greater), the test results of cemented coarse-gained soil parameters (The maximum size of the gravel grains is 20 mm), and cemented poorly graded sand–gravel mixture parameters (The maximum size of the gravel grains is 12.5mm) [13,15,27-28]. The type, chemical and physical properties of the raw materials included cement and coarse and fine aggregates of those are different from those used in this study.The fitting functions, i.e., Equations (3), (4), (8), (9), (10), (11), (15) and (16) for the parameters in Equations (5), (12), (13), (17), and (22) under different cement contents can be used to fit the test results of CSG materials parameters by Yang and Fu et al., the test results of cemented coarse-gained soil parameters by Li in Figure 16[13,27-28]. These verification results demonstrate that the expressions for quantified indexes of the deformation characteristics can also be used to well describe the deformation characteristics of other types of CSG materials and cemented coarse-gained soil.

The expressions for the deformation properties of CSG material are used to simulate the deformation characteristics of a cemented poorly graded sand–gravel mixtureas shown in Figure 17 [15]. However, some expressions considering the effect of confining pressure can not fit the corresponding test results for the material well. The main reason could be that the grading and size of the aggregate particles, cement content, and confining pressure of the cemented poorly graded sand–gravel mixture are different with those of the cemented granular material described above.

In particular, research on the effect of confining pressure on the deformation properties of some special concrete material has rarely been considered and thus these expressions in the study are not suitable[21-22]. These expressions including Equations (5), (12), (13), (17), and (22) in this study can be considered to be applicable to the cemented granular materials with a cement content less than 100 kg/m3 and a maximum particle size of the aggregate greater than 20 mm.

4. Conclusions

5) The expressions in this study are applicable when determining deformation properties of cemented granular materials with a cement content less than 100 kg/m3 and a maximum particle size of the aggregates greater than 20 mm under different confining pressures and cement contents.

Reviewer 2 Report

The authors entirely reviewed the article, which has been strongly improved. Now the introduction clearly presents the literature of the topic and the data analysis can be easily followed. Nevertheless, I have some further suggestions for the authors.

In general, the result and discussion section has been enriched, but it can be further improved. I would suggest merging the sections 3.1.2 and 3.2.1, since they refer to the same relations and lead to the same conclusion (point 3 of the conclusions). The discussion around the peak volumetric strain, axial strain and failure strain could be grouped for the sake of clarity. As a matter of fact, equations 6, 10and 14 refer to same concept, such as Figures 8, 11 and 14; eq.7, 11 and 15;  eq. 8, 12 and 16; eq. 9, 13 and 17, Fig. 9, 12 and 15; Fig. 10, 13, 16.

Concerning the new section 3.3, it was interesting to see how the equations here derived well describe also other materials. The discussion around this section (lines 557-570) should be improved to help the reader appreciate this result. Last but not least, if no error bars are shown in the figures, it is not possible for the reader to see if the expressions well fit the experimental results.

I include here below some more specific comments:

- According to the Instructions for Authors, the article should include the following sections: Introduction, Materials and Methods, Results, Discussion, Conclusions (optional). Thus, the section 2 should be renamed as “Materials and Methods”.

- English language: at lines 16-20 the sentence is very long and the part “on the properties of CSG material” does not seems appropriately placed to me; at lines 411-412 and 566-569 the sentences are not clear.

- Check the style at line 371

- Lines 106-117: refer to the citations in the right place

- Line 373: “the slope of point 1” is misleading, a point does not have a slope

- Line 386 and 394: you need to describe E01 and n parameters. They are dimensionless parameter, but what do they describe? What are they related to?

- Fig 5, 6, 9, 10, 12, 13, 15, 16 : in the legend you can see “test values, fitted curves, and the value for the experimental verification”. You should introduce these values and, especially, explain the use the ‘value for the experimental verification’.

- As already pointed out, error bars are particularly important when fitting some experimental data with models. In the author's reply you answered that the problem was solved, but the error bars are still missing. As an example to appreciate the importance of them, at lines 566-568 you report that the expression does not fit well the experimental values, but you cannot state this without considering the error bars. 

Author Response

Point 1: In general, the result and discussion section has been enriched, but it can be further improved. I would suggest merging the sections 3.1.2 and 3.2.1, since they refer to the same relations and lead to the same conclusion (point 3 of the conclusions). The discussion around the peak volumetric strain, axial strain and failure strain could be grouped for the sake of clarity. As a matter of fact, equations 6, 10and 14 refer to same concept, such as Figures 8, 11 and 14; eq.7, 11 and 15;  eq. 8, 12 and 16; eq. 9, 13 and 17, Fig. 9, 12 and 15; Fig. 10, 13, 16.

Response 1: This section has been modified and is as follows:

3.1. Stress–strain and volumetric strain–axial strain curves

The stress–strain and strength characteristics in Figures 1(a), 2(a), 3(a), and 4(a) have been introduced by Yang et al., and these sections will not be discussed here [14]. The slope of the stress–strain curve when the axial strain is 0, called the initial modulus herein, and the strain at point 2 (the peak point of the stress–strain curve), called the failure strain, are key issues in the deformation calculation of CSG dams. Therefore, the effects of various cement contents and confining pressures on the initial modulus and failure strain are studied in this study.

It can be seen from Figures 1(b), 2(b), 3(b), and 4(b) that the volumetric strain–axial strain curves of the CSG material before the peak volumetric strain show obvious nonlinearity. When the cement content is low, the shear shrinkage and dilatancy of the CSG material are similar to those of the rockfill material; when the cement content is high, the dilatancy is more obvious, and the initial slopes of the volumetric strain–axial strain curves are less affected by the confining pressure. When the cement content is approximately 100 kg/m3, the confining pressure has little influence on the initial slope of the CSG material. The volumetric strain and axial strain at point 3 (the peak point of the volumetric strain–axial strain curve) decrease with an increase in the cement content. The volumetric strain–axial strain characteristics of the CSG material above are roughly the same as those of polymer rockfill materials, sand reinforced with fibers, etc. [16,24].

Because the CSG material begins to exhibit dilatancy at point 3, point 3 is also called the initial dilatancy point. The slope of the volumetric strain–axial strain curve when the axial strain is 0 (point 4) is called the initial tangent volumetric ratio here. These are also key quantities that are considered in the deformation calculation of CSG dams and quantitatively described below.

3.2. Initial modulus

Figure 5 illustrates the initial modulus under varying confining pressures and cement contents. As shown in the figure, the initial modulus increases exponentially with an increase in confining pressure under a certain cement content. This is consistent with the observations made by previous studies on other types of CSG materials [10-12]. The relationship between the initial modulus and confining pressure of CSG material adopted by some of these studies is as follows [10]:

                                              ,                                (1)

where E01 and n are dimensionless parameters related to the type of the soil, rockfill material, etc.; the atmospheric pressure Pa is 100 kPa; E01Pa represents the initial modulus when the confining pressure is 100 Pa; and n represents the growth index of the initial modulus. Through a regression analysis of data from Figure 5 and Equation (1), the values of E01 and n are obtained and are shown in Table 4. Values of Correlation coefficient R2 in the table that are greater than 0.97 indicate that the calculated results of Equation (1) fit the experimental results well. However, when the confining pressure is 0, the initial modulus is 0 in Equation (1), and this is inconsistent with the actual value of the modulus. Thus, the relationship between the initial modulus and confining pressure for a CSG material, determined by Cai et al. and Fu et al., is expressed as follows [11,13]:

,                                (2)

where E0 and n are dimensionless parameters; E0Pa represents the initial modulus of the CSG material when the confining pressure is 0; n represents the growth index of the initial modulus. Through a regression analysis of data from Figure 6 and Equation (2), the values of E0 and n are obtained and are shown in Table 5. Values of R2  in the table that are greater than 0.98 indicate that the calculated results of Equation (2) fit the experimental results well, thereby demonstrating that Equation (2) can be used to describe the initial modulus of CSG material as a function of the confining pressure.

The test values of E0 and n under different cement contents are also shown in Figures 6 and 7, respectively. As can be seen from Figure 6, when Cc is close to 0, E0 is close to the corresponding value of rockfill material. With an increase of Cc, E0 increases. The parameter n is close to the corresponding value of rockfill material in Figure 7, when Cc is close to 0. And with an increase of Cc, n decreases and is close to 0.

Figure 5. The relationship between the initial modulus Ei and confining pressure σ3 under varying cement contents

Table 4  Values of R2, E01 and n

Cc(kg/m3)

Formula

Correlation coefficient R2

E01

n

20

Ei= 86078(σ3/100)0.51

0.98

860.78

0.51

40

Ei= 09766(σ3/100)0.43

0.99

1097.66

0.43

60

Ei = 13587(σ3/100)0.35

0.99

2135.87

0.35

80

Ei = 66396(σ3/100)0.29

0.99

3663.96

0.29

100

Ei = 33524(σ3/100)0.16

0.97

8335.24

0.16

Table 5  Values of R2, E0 and n for Equation (2)

Cc

(kg/m3)

Formula

Correlation   coefficient R2

E0

n

20

Ei = 65651(1+σ3/100)0.60

0.99

656.51

0.60

40

Ei =109766(1+σ3/100)0.45

0.99

1097.66

0.45

60

Ei=178000(1+σ3/100)0.41

0.99

1780.00

0.41

80

Ei =315000(1+σ3/100)0.31

0.99

3150.00

0.31

100

Ei = 06177(1+σ3/100)0.20

0.98

7061.77

0.20

Figure 6. The relationship between the parameter E0 and cement content Cc

Figure 7. The relationship between the parameter n and cement content Cc

In the study, E0 and n were fitted with the corresponding parameters of CSG material for cement contents of 20, 40, 80, and 100 kg/m3; we thus get

,                                (3)

,                                (4)

where E0 and b are the fitting coefficients of the relationship between E0 and Cc; and c are the fitting coefficients of the relationship between n and Cc; Cc0 is reference cement content with a value of 1 kg/m3. The curves fitted by Equations (3) and (4) are shown in Figures 6 and 7, respectively. Values of R2 in these curves that are greater than 0.97 indicate that the calculated results of Equations (3) and (4) can fit the experimental results well.

From Equations (2), (3), and (4), we can derive the relationship of the initial modulus of CSG material with the cement content and confining pressure as follows:

,                              (5)

Equation (5) can quantitatively reflect the influence of confining pressure and cement content on the initial modulus Ei : When the cement content Cc is 0, this equation can be used to describe the relationship between the initial modulus of the rockfill material and confining pressure; with anincrease in the cement content, Ei under a certain confining pressure increases; when the cement content increases to 80 kg/m3, Ei is less affected by the confining pressure.

3.3. Axial strain at the peak points of the test curves

In this study, the axial strain at the peak points of the stress–strain curves represents the failure strain, and the axial strain at the peak points of the volumetric strain–axial strain curves represents the axial strain at the initial dilatancy point. Figure 8 shows that the values of these axial strain decrease with increasing cement content and decreasing confining pressure. However, the value of failure strain is larger than that of the axial strain at the initial dilatancy point under the same cement content and confining pressure. This suggests that the dilatancy is usually observed before failure of the CSG material. When the cement content is higher than 60 kg/m3, the failure strain is almost the same and in the range of 1.2% to 1.5% under a certain confining pressure, which indicates that the brittleness of CSG material is more obvious with increasing cement content. The axial strain at the initial dilatancy point of CSG material with a certain cement content is also almost the same and in the range of 0.9% and 1.3%. According to the analysis in Figure 8, the relationship between failure strain εm and confining pressure and the relationship between axial strain at the initial dilatancy point εn and confining pressure can be expressed as follows:

,                                (6)

,                                (7)

where λ0 and λ1 are dimensionless parameters related to the cement content and the slope of those lines; d0 and d1 are the vertical intercept and dimensionless parameters related to the cement content. Through a regression analysis of data from Figure 8, the values of E01 and n in Equation (6) are shown in Table 6, and the values of E0 and n in Equation (7) can be obtained and shown in Table 7. Values of R2  in those tables that are greater than 0.95 indicate that the calculated results of Equations (6) and (7) can fit the experimental results well.

By fitting the test values of λ0 and d0 for different cement contents, which are shown in Figure 9 and Figure 10 of CSG material with the cement contents of 20, 40, 80, and 100 kg/m3, we have

,                                 (8)

,                                (9)

where a0 and c0 are the fitting coefficients of the relationship between λ0 and Cc; l0, m0, and n0 are the fitting coefficients of the relationship between d0 and Cc. The curves fitted by Equations (8) and (9) are shown in Figures 9 and 10, respectively. Values of R2 that are greater than 0.99 indicate that the calculated results of Equations (8) and (9) fit the experimental results well.

  The test values of λ1 and d1 under different cement contents are shown in Figures 11 and 12, respectively. By fitting the data of the triaxial shear test with the cement contents of 20, 40, 80, and 100 kg/m3, we have

,                                (10)

,                              (11)

where a1 and c1 are the fitting coefficients of the relationship between λ1 and Cc; l1, m1, and n1 are the fitting coefficients of the relationship between d1 and Cc. The curves fitted by Equations (10) and (11) are shown in Figures 12 and 13, respectively. Values of R2 that are greater than 0.98 indicate that the calculated results of Equations (10) and (11) fit the experimental results well.

By substituting Equations (8) and (9) into Equation (6) and Equations (10) and (11) into Equation (7), we can derive the relationship between failure strain εm and cement content Cc and confining pressure, and the relationship between the axial strain at the initial dilatancy point εn and the cement content Cc and confining pressure as follows:

,                            (12)

.                            (13)

(a)                             (b)

Figure 8. The relationship between the axial strain and confining pressure σ3:(a) the failure strain εm;; the axial strain at the initial dilatancy point εn

Table 6 Values of R2, λ0 and d0

Cc(kg/m3)

Formula

Correlation coefficient R2

λ0 (%)

d0 (%)

20

εm=0.33(σ3/100)+4.7

0.98

0.33

4.7

40

εm=0.20(σ3/100)+1.92

0.99

0.2

1.92

60

εm=0.065(σ3/100)+1.22

0.96

0.065

1.22

80

εm=0.058(σ3/100)+0.99

0.98

0.058

0.99

100

εm=0.041(σ3/100)+0.96

0.97

0.041

0.96

Table 7 Values of R2, λ1 and d1

Cc(kg/m3)

Formula

Correlation   coefficient R2

λ1(%)

d1(%)

20

εn=0.36(σ3/100)+2.29

0.97

0.36

2.29

40

εn=0.23(σ3/100)+1.23

0.99

0.23

1.23

60

εn=0.082(σ3/100)+0.84

0.95

0.082

0.84

80

εn=0.053(σ3/100)+0.72

0.96

0.053

0.72

100

εn=0.041(σ3/100)+0.65

0.97

0.041

0.65

Figure 9. The relationship between the parameter λ0 and cement content Cc

Figure 10. The relationship of the parameter d0 and cement content Cc

Figure 11. The relationship between parameter λ1 and cement content Cc

Figure 12. The relationship of parameter d1 and cement content Cc

Point 2:Concerning the new section 3.3, it was interesting to see how the equations here derived well describe also other materials. The discussion around this section (lines 557-570) should be improved to help the reader appreciate this result. Last but not least, if no error bars are shown in the figures, it is not possible for the reader to see if the expressions well fit the experimental results.

Response 2: This section has been modified and is as follows:

The expressions for the deformation properties of CSG material are used to simulate the deformation characteristics of a cemented poorly graded sand–gravel mixtureas shown in Figure 17 [15]. However, some expressions considering the effect of confining pressure can not fit the corresponding test results for the material well. The main reason could be that the grading and size of the aggregate particles, cement content, and confining pressure of the cemented poorly graded sand–gravel mixture are different with those of the cemented granular material described above.

(a)                                       (b)

(c)

Figure 17. The verification of the expressions for the deformation characteristics of the cemented poorly graded sand–gravel mixture under different cement contents: (a) the relationship between the failure strain εm and confining pressure σ3 ; (b) the relationship between the axial strain of the peak point εn and confining pressure σ3; (c) the relationship between the peak volumetric strain εvd and confining pressure σ3.

Point 3: According to the Instructions for Authors, the article should include the following sections: Introduction, Materials and Methods, Results, Discussion, Conclusions (optional). Thus, the section 2 should be renamed as “Materials and Methods”.

Response 3:This section has been modified.

Point 4:English language: at lines 16-20 the sentence is very long and the part “on the properties of CSG material” does not seems appropriately placed to me; at lines 411-412 and 566-569 the sentences are not clear.

Response 4: The sentence at lines 16-20 has been modified and is as follows: Knowing the deformation properties of cemented sand and gravel (CSG) material is can helpful tohelp construct reasonable constitutive models for the material, which can be used to simulate the structural performance of various practical projects including CSG dams. In this study, to investigate the effect of cement content on the deformation properties of CSG material, we employ triaxial compressive tests for cement contents of 20, 40, 60, 80, and 100 kg/m3 with a confining pressure range of 0.3–1.2MPa, and theoretically analyze the results by the regression analysis prediction method.

Point 5:at lines 411-412 and 566-569 the sentences are not clear

Response 5:  All sentences in this manuscript  have been retouched.

Point 6: Check the style at line 371

Response 6: The style at line 371 has been checked and is as follows:

,                                                            (18)

Point 7: Lines 106-117: refer to the citations in the right place

Response 7:This section at lines 106-117 has been modified in Round 1. Citations in revised manuscripts are subject  to the journal requirements.

Point 8: Line 373: “the slope of point 1” is misleading, a point does not have a slope

Response 8: “the slope of point 1” is converted into “The slope of the stress–strain curve when the axial strain is 0”.

Point 9:Line 386 and 394: you need to describe E01 and n parameters. They are dimensionless parameter, but what do they describe? What are they related to?

Response 9: This section has been modified and is as follows:

Where E01 and n are dimensionless parameters related to the type of the soil, rockfill material, etc.; the atmospheric pressure Pa is 100 kPa; E01Pa represents the initial modulus when the confining pressure is 100 Pa; and n represents the growth index of the initial modulus.

Point 10: Fig 5, 6, 9, 10, 12, 13, 15, 16 : in the legend you can see “test values, fitted curves, and the value for the experimental verification”. You should introduce these values and, especially, explain the use the ‘value for the experimental verification’.

Response 10:This section has been modified and is as follows:

The parameters of the deformation properties for CSG material with the cement contents of 20, 40, 80, and 100 kg/m3 are test values, and the calculated values of the predictive models form the fitted curves as shown  in Figures 6, 7, 9, 10, 11, 12, 14, and 15.

The experimental results of CSG material with the cement content of 60 kg/m3 for the experimental verification, and thier calculated results are also shown in Figures 6, 7, 9, 10, 11, 12, 14, and 15.

Point 11: As already pointed out, error bars are particularly important when fitting some experimental data with models. In the author's reply you answered that the problem was solved, but the error bars are still missing. As an example to appreciate the importance of them, at lines 566-568 you report that the expression does not fit well the experimental values, but you cannot state this without considering the error bars. 

Response 11: This section has been modified and is shown in tables 3-8 and Figures 16-17.

Round 3

Reviewer 1 Report

The paper can be accepted.